# The Evolving Role of CD8^+^CD28^−^ Immunosenescent T Cells in Cancer Immunology

**DOI:** 10.3390/ijms20112810

**Published:** 2019-06-08

**Authors:** Wei X. Huff, Jae Hyun Kwon, Mario Henriquez, Kaleigh Fetcko, Mahua Dey

**Affiliations:** 1Department of Neurosurgery, Indiana University School of Medicine, Indianapolis, IN 46202, USA; wxia@iupui.edu (W.X.H.); kwonjaeh@iu.edu (J.H.K.); mhenriqu@iu.edu (M.H.); 2Department of Neurology, University of Illinois at Chicago School of Medicine, Chicago, IL 60612, USA; kaleighfetcko@gmail.com

**Keywords:** CD8^+^CD28^−^ T cells, cancer immunology, glioblastoma, immunotherapy, malignant glioma, cancer

## Abstract

Functional, tumor-specific CD8^+^ cytotoxic T lymphocytes drive the adaptive immune response to cancer. Thus, induction of their activity is the ultimate aim of all immunotherapies. Success of anti-tumor immunotherapy is precluded by marked immunosuppression in the tumor microenvironment (TME) leading to CD8^+^ effector T cell dysfunction. Among the many facets of CD8^+^ T cell dysfunction that have been recognized—tolerance, anergy, exhaustion, and senescence—CD8^+^ T cell senescence is incompletely understood. Naïve CD8^+^ T cells require three essential signals for activation, differentiation, and survival through T-cell receptor, costimulatory receptors, and cytokine receptors. Downregulation of costimulatory molecule CD28 is a hallmark of senescent T cells and increased CD8^+^CD28^−^ senescent populations with heterogeneous roles have been observed in multiple solid and hematogenous tumors. T cell senescence can be induced by several factors including aging, telomere damage, tumor-associated stress, and regulatory T (Treg) cells. Tumor-induced T cell senescence is yet another mechanism that enables tumor cell resistance to immunotherapy. In this paper, we provide a comprehensive overview of CD8^+^CD28^−^ senescent T cell population, their origin, their function in immunology and pathologic conditions, including TME and their implication for immunotherapy. Further characterization and investigation into this subset of CD8^+^ T cells could improve the efficacy of future anti-tumor immunotherapy.

## 1. Introduction

The conflict between cancer and the immune system has long been established [1]. Immunotherapies are being investigated to augment the anti-tumor effects of the immune system and promote long-term cancer control [2]. CD8^+^ cytotoxic T cells (CTLs) are the main players driving the adaptive immune response against cancer and execute tumor-specific immune responses, rendering them the primary endpoint to most immunotherapies [3,4]. Establishment of effective antigen-specific CD8^+^ T cells enabled preliminary clinical success of cancer vaccines, oncolytic viruses, adoptive cellular therapy, and checkpoint inhibitors in several cancers including melanoma, lung cancer, renal cell cancer, Hodgkin’s lymphoma, etc. [5,6,7]. Unfortunately, despite their promise, the efficacy of these treatments varies depending on the type and location of the tumor, and has been ineffective in poorly immunogenic cancers such as glioblastoma (GBM) [7,8,9,10,11,12,13].

A variety of T cell deficiencies have been identified in immunosuppressive tumors that contribute to the ultimate ineffectiveness of CD8^+^ CTL-mediated tumor killing [14]. Immune tolerance, anergy, and exhaustion of CD8^+^ T cells have been studied extensively in the past [14,15,16,17]. While the concept of immune senescence, defined by terminally differentiated cells in cell cycle arrest after extensive replication or in response to cellular damage or stress, has been well established with aging and chronic infections [18,19,20,21], our knowledge of its role in cancer is still in its early stages. CD28 is an indispensable costimulatory molecule needed for the activation of T cells and its role is critical to the proper activation of CD8^+^ CTLs [22]. Current evidence shows that the downregulation of CD28 is a hallmark of senescent CD8^+^ T cells and CD28^−^ senescent T cells display immunosuppressive functions in cancer [23,24,25,26].

In this review, we will focus on the recent advances in our understanding of CD8^+^CD28^−^ T cells. We first will discuss cellular senescence and the evolution of the CD28^−^ T cells. Next, we will review the significance of CD8^+^CD28^−^ T cells in multiple disease processes, including transplant, autoimmune disease, chronic viral infection, and cancer, including CNS tumors. Finally, we will discuss the functional implications of CD28^−^ T cells in onco-immunology and the important areas of future investigation on novel immunotherapeutic strategies.

## 2. Role of CD8^+^ T Cells in Cancer Immunology

CD8^+^ T cells are a subset of lymphocytes committed to detecting peptide antigens presented by major histocompatibility complex (MHC) class I molecules (Figure 1) [27,28]. CD8^+^ T cells arise from common lymphoid progenitors that migrate from the bone marrow to the thymus where they pass through a series of distinct phases of maturation [29,30]. The naïve CD8^+^ T cell pool is comprised of polyclonal T cells that express CD28, CCR7, and CD62L, the latter two allowing them to recirculate between blood and secondary lymphoid organs [31,32]. Initial priming of CD8^+^ T cells involves T cell receptor (TCR) recognition of peptide/MHC complexes presented by professional antigen presenting cells (APCs), such as dendritic cells (DCs). DCs also express surface markers CD70 and CD80/CD86 for binding to CD27 and CD28 receptors expressed on CD8^+^T cells. This provides a critical secondary signal for T cell activation. Host cells, including cancer cells, can serve as targets for previously activated CD8^+^ T cells by processing and presenting antigenic tumor peptides by MHC class I.

In addition, CD4^+^ T helper cells interact with CD8^+^ T cells and modulate CD8^+^ T cell activation [34,35,36]. Activated CD4^+^ T helper cells can secrete a variety of cytokines, such as interferon-gamma (IFN-γ) and IL-2, and facilitate CD8^+^ T cells’ optimal proliferation and activation [28,37]. CD4^+^ T cells could also help with DC maturation for expression of costimulatory molecules and secretion of cytokines that contribute to CD8^+^ T cell priming [38]. A similar mechanism is also carried out by natural killer (NK) cells, showing that there is collaboration between CD4^+^ T cells with NK and DCs for induction of CD8^+^ T cell priming [28,38].

Upon activation, effector CD8^+^ CTLs destroy antigen-expressing cancer cells primarily utilizing two main pathways: granule exocytosis (such as perforin and granzyme) and death ligand/receptor-mediated apoptosis (such as Fas ligand and TRAIL) [39]. Additionally, activated CD8^+^ T cells release IFN-γ and tumor necrosis factor alpha (TNF-α) to induce cytotoxicity in the target cells and stimulate M1 macrophage-mediated anti-tumor response [28]. In multiple solid tumors, tumor-infiltrating CD8^+^ CTLs can be used as a prognostic factor [40,41,42,43,44,45,46,47,48]. For example, in breast cancer, significantly increased CD8^+^ T cells at tumor sites have been shown to have an inverse correlation with advanced tumor stages and a positive correlation with clinical outcomes [41,49,50]. Similar findings of a favorable prognosis associated with the accumulation of tumor-infiltrating CD8^+^ T cells were reported in colorectal, oral squamous cell, pancreatic, and ovarian carcinomas [43,44,45,47,48,51].

## 3. CD28 Costimulatory Receptor

The CD28 costimulatory receptor, a 44-kDa membrane glycoprotein, is expressed on nearly all human T lymphocytes at birth [52]. Binding of the CD28 receptor on T cells provides an essential second signal alongside TCR ligation for naïve T cell activation. CD28 signaling has diverse effects on T cell function, including orchestrating membrane raft trapping at the immunological synapse, transcriptional changes, downstream post-translational modifications, and actin cytoskeletal remodeling [52,53,54]. This leads to intracellular biochemical events such as survival and proliferation signals, induction of IL-2, activation of telomerase, stabilization of mRNA for several cytokines, increased glucose metabolism, and enhanced T cell migration and homing [52,55,56].

CD28 family of costimulatory molecules also includes ICOS, CTLA-4, PD-1, PD1H, TIGIT, and BTLA [41]. This family of receptors and ligands has considerable complexity in both binding pattern and biological effects. For instance, CD28 (activating) and CTLA-4 (inhibitory) are highly homologous and compete for the same ligands (CD80 and CD86) and regulate immune response by providing opposing effects [51,52].

The critical role of CD28 in induction of immune response was demonstrated in mice treated with CD28 antagonist, which induced antigen specific tolerance and prevented the progression of autoimmune diseases and organ graft rejection [57]. This has led to the development of abatacept (Orencia^®^ Bristol-Myers Squibb, New York, NY, USA) [58] and belatacept (Nulojix^®^ Bristol-Myers Squibb New York, NY, USA) [59], a modified antibody composed of Fc region of IgG1 fused to the extracellular domain of CTLA-4, which bind to CD80/86 on APCs and block the costimulatory signaling by CD28. Abatacept and belatacept are used clinically to treat rheumatoid arthritis and organ transplant rejection, respectively [56,60]. 

On the other hand, the use CD28 agonists to awaken T cells from the tolerant state could lead to new therapies to re-activate the immune system for the treatment of infectious disease [61] and cancer [62]. Although, in a phase I trial of systemic administration of CD28 superagonist monoclonal antibodies (mAb) (TGN1412), uncontrolled CD28 signaling led to a potent induction of downstream immune activation independent of TCR-CD3 complex resulting in catastrophic systemic inflammatory syndromes in six volunteer subjects [63]. Investigation of these unexpected serious adverse events have led to better design of clinical trials and appreciation of differences in CD28 expression and regulation between species (critical for transitioning preclinical testing to clinical investigations) [64,65]. An updated CD28 superagonist TAB08 is under clinical testing for rheumatoid arthritis [64]. In addition, localized or targeted use of anti-CD28 mAbs has much potential such as incorporating the intracellular costimulatory domain of CD28 into CAR (chimeric antigen receptor) T cells for adoptive transfer immunotherapy and the use of CD28 agonist aptamer with tumor vaccine [66,67,68].

Importantly, the use of these therapeutics are in clinical trials for a variety of disease states including solid neoplasms and inflammatory diseases (Table 1). Although previous experience with CD28 agonist mAbs has been disappointing, headway is being made in their use in solid tumors and rheumatoid arthritis. Perhaps out of an abundance of caution, current clinical trials for the use of CD28 agonists are testing their safety, efficacy, and tolerability in patients and are undergoing dose escalation studies. Fortunately, significant progress has been made into CAR-T cell therapy incorporating costimulatory domains and has led to the FDA-approval of the CAR-T cell therapy tisagenlecleucel (KYMRIAH^®^ Novartis Pharmaceuticals, Basel, Switzerland) for relapse or refractory acute lymphoblastic leukemia patients [69,70].

Furthermore, recent progress in the manipulation of other costimulatory molecule such as ICOS, CD137 (4-1BB), OX40, and GITR has also demonstrated tremendous therapeutic potential [74]. Activation of ICOS, CD137, and OX40 via mAbs and aptamers improved T cell proliferation, function, and overall antitumor response [74,75,76,77]. Targeting of GITR exhibited effects on both effector and regulatory T cells. Ligation of constitutively expressed GITR on regulatory T cells caused depletion in their number, loss lineage stability and immunosuppressive function [78,79], while GITR agonists work synergistically with PD-1 blockage to promote CD4 and CD8 accumulation in murine ovarian cancer [80]. Blockade of inhibitory receptor CTLA-4 have been shown to be effective in enhancing CD28 signaling and augmenting ICOS stimulation [74,81]. Combination of checkpoint inhibitors and costimulatory agonists presents an exciting avenue of cancer treatment and several clinical studies are currently investigating their benefit [74].

## 4. Cellular Senescence in the Immune System

Cellular senescence is a state of cell cycle arrest in respond to cellular damage or stress to prevent neoplastic transformation [82]. Cellular senescence have been identified in areas of physiological homeostasis, such as development [82], wound healing [83], and placental natural killer lymphocytes [84]. However, cellular senescence also contributes to the loss of function associated with aging and age-related disease as well as chronic viral infection, neurodegenerative disease, and cancer [18,85,86]. Two categories of cellular senescence have been described in literature: the first is aging associated, telomere-dependent replicative senescence and the second is stress-induced premature senescence, also known as telomere-independent senescence [82,87]. Oncogene-induced senescence is one of the well-described mechanisms for premature senescence [87,88]. Regardless of the initiating mechanism, cells that undergo senescence survive by exhibiting a variety of phenotypical and molecular features (Figure 2), including morphological changes, cell division blockage, change of sensitivity against apoptosis, metabolic dysfunction, and a specialized secretory activity termed senescence associated secretory phenotype (SASP) [20]. Additional characteristics include nuclear p16 and p21 expression [89,90,91], DNA damage [92], senescence associated heterochromatin foci (SAHF) [93], and increased lysosomal senescence-associated β-galactosidase (SA-β-gal) activity [91]. Recently, lipofuscin accumulation was also established as a hallmark of senescent cells [94].

Cellular senescence also occurs in the human immune system [18,95]. The effectiveness of the immune response declines with age particularly in the elderly population [96,97]. Immune deficiencies start to appear in DCs, NK cells, and monocytes/macrophages with aging, and it was proposed that myeloid-derived suppressor cells (MDSC) could also induce senescence in T and B cells compartment in diverse inflammatory conditions [96,97]. Importantly, lymphocytes, especially T cells, show the most considerable changes with aging [95,98]. Among the various complex features that contribute to the aging-associated changes in T cell immunity, the accumulation of CD28^−^ T cells is one of the hallmark phenomena in T cell immunosenescence [26,99,100]. TME can also induce senescence in tumor-infiltrating T cells [14].

## 5. Role of CD8^+^CD28^−^ T Cells

Although CD28 is expressed on the majority of the CD8^+^ T cells at birth [52], normal aging process and activation of CD8^+^ T cells invariably leads to CD28 downregulation [101]. CD8^+^CD28^−^ cells represent a distinct population distinguishable from the general population of CD8^+^CD28^+^ T cells [99], which are known for their crucial role in the clearance of cancer and intracellularly infected cells, in terms of their phenotype and function [102]. *In vitro* studies showed purified CD28^+^ T cells progressively lose CD28 during each successful stimulation, with the CD8^+^ T cells losing their CD28 more rapidly than the CD4^+^ T cells [26,103,104]. The differential rate of CD28 loss is associated with the rapid inactivation of telomerase and CD8^+^ T cells reach replicative senescence faster than CD4^+^ T cells, at which stage T cells are no longer able to enter mitosis but still remain viable [105]. Thus, these CD8^+^CD28^−^ T cells are defined as senescent T cells. Less than 50% of the CD8^+^ T cell compartment of elderly or chronically infected individuals are CD28^+^ while up to 80% of CD4^+^ T cells maintain their CD28 expression even in the centenarians [26,103]. Interestingly, a large proportion of CD8^+^CD28^−^ T cells of elderly persons also have lower levels of CD8 expression [106,107]. Although the significance of this observation is unknown, downregulation of the expression of CD8 and CD4 molecules is characteristic for activated T cells, suggesting that those CD8^low^CD28^−^ T cells subset represent senescent lymphocytes that are chronically activated from either common persistent antigens (in the setting of aging) or persistent infection or inflammation (in the setting of cancer) [25,108].

## 6. Characteristics of CD8^+^CD28^−^ Senescent T cells

CD8^+^CD28^−^ T cells are highly oligoclonal and terminally differentiated effector lymphocytes that have lost their capacity to undergo cell division [23,108]. They are functionally heterogeneous and their characteristics vary depending on the context where they are found (Figure 3) [23,108]. They also express a variety of other NK cell-related receptors including KIR, NKG2D, CD56, CD57, CD94, and Fc-γ receptor IIIa and have features crossing the border between innate and adaptive immunity [109,110]. Alterations in the costimulatory receptor NKG2D signaling and expression levels in CD8^+^ T cells can lead to autoimmune conditions that are either TCR dependent or TCR-independent [111,112,113]. Gained expression of CD57, also known as HNK-1 (human natural killer-1), is a common feature associated with circulating senescent T cells, and increased CD8^+^CD28^−^CD57^+^ senescent T cells were identified in multiple pathological conditions, including HIV infection, multiple myeloma, lung cancer, and chronic inflammation conditions such as diabetes and obesity [99,114,115]. Although expression of CD57 is linked to antigen-induced apoptosis of CD8^+^ T cells [116], the acquisition of CD94 has been reported to confer resistance to apoptosis in CD8^+^CD28^−^ T cells. [117] Similarly, CD8^+^CD28^−^ T cells are often associated with the lack of perforin, rendering them ineffective Ag-specific killers in chronic viral infections [21,118,119,120]. On the other hand, in certain disease processes such as chronic obstructive pulmonary disease (COPD) and rheumatoid arthritis, they have been reported to express increased levels of cytotoxic mediators, perforin and granzyme B, and pro-inflammatory cytokines, IFN-γ and TNFα, where CD8^+^CD28^−^ T cells can cause significant damages to normal surrounding tissue in an antigen-nonspecific manner [121].

CD8^+^CD28^−^ T cells are also shown to be immunosuppressive and function as regulatory T cells [122,123,124,125]. For example, CD8^+^CD28^−^ T cells directly inhibit Ag-presenting function of DCs by inducing inhibitory receptors, such as immunoglobulin like transcript 3 (ILT3) and ILT4, which leads DCs to be immune tolerant than immunogenic [122,126]. Such tolerogenic DCs anergize alloreactive CD4^+^CD25^+^ T cells and convert them into regulatory T cells, which in turn, continue the immunosuppressive cascade by tolerizing other DCs and amplify T cell immunosenescence [126,127]. *In vivo*, CD8^+^CD28^−^ T cells have been directly correlated with the suppression of antigen-specific T cell responses in patients with plasma cell dyscrasia [123]. Therefore, their characteristics and functions in immunity range from reduced antigen-specific killing to enhanced cytotoxic abilities and from crossing innate immunity function to promoting immune regulation.

## 7. CD8^+^CD28^−^ T cells in Pathologic Conditions

CD8^+^CD28^−^ T cells play a significant role in pathological conditions [18,99,100,121]. High populations of these cells have been associated with chronic viral infections including human immunodeficiency virus (HIV), hepatitis C virus, cytomegalovirus (CMV), and human parvovirus B19 [99]. Shortened telomeres, reduced IL-2 production, and increased IL-6 production were observed in these cells [19]. The loss of CD28 also serves as a prognostic indicator for viral infection. For instance, increased frequency of CD8^+^CD28^−^ T cells in the early stage of HIV infection correlates with faster progression to AIDS [99]. Additionally, higher levels of CD8^+^CD28^−^ T cells are associated with subclinical carotid artery disease in HIV-infected women [99]. Older CMV seropositive individuals, who had higher number of CD8^+^CD28^−^ cells, responded poorly to vaccines and had early mortality compared to aged-matched CMV seronegative counterparts [19].

A heterogeneous role was reported for CD8^+^CD28^−^ T cells in autoimmune diseases [99,108]. Senescent T cell population is increased in patients with Grave’s disease and ankylosing spondylitis and these cells’ cytotoxicity contributes to autoimmune response [128,129]. In rheumatoid arthritis patients, clinical response to abatacept, a CD80/86-CD28 T cell co-stimulation modulator, is associated with a concomitant decrease in CD8^+^CD28^−^ T cells, suggesting prognostic value for this phenotype [56]. In contrast, patients with systemic lupus erythematosus were found to have reduced CD8^+^CD28^−^ T cells [130]. Similarly, in a mice model for multiple sclerosis, adoptive transfer of CD8^+^CD28^−^ regulatory T cells have been shown to prevent autoimmune encephalomyelitis [131]. Though treated as a single phenotype, CD8^+^CD28^−^ T cells represent a heterogeneous group that has differential activities in different pathologic conditions. In solid organ transplant recipients, CD8^+^CD28^−^ T cells have been found to undergo oligoclonal expansion and play a suppressive role and promote allograft tolerance [108,132]. Elevated CD8^+^CD28^−^ T cell population in liver transplant patients are associated with better graft function and reduced rejection rates [133] and contribute to reducing immunosuppressant dosage [124]. In addition, the presence of CD8^+^CD28^−^ T cells is associated with decreased CD80/86 expression and increased inhibitory receptor (ILT3, ILT4) in circulating APCs, implying an immunosuppressive role of this subset [126,133].

## 8. CD8^+^CD28^−^ T cells and Cancer

The cycle of anti-tumor immunity starts with the presentation of cancer antigens released from cancer cell turnover. Resident tissue DCs or lymph nodes residing DCs capture cancer antigens and present the antigens in the form of peptide-MHC I complex to activate naïve CD8^+^ T cells. Activated effector CD8^+^ T cells travel through blood and lymphatic to reach tumor beds where they execute cancer-specific killing. This leads to further endogenous antigen release and DC activation, thereby closing the cycle for anti-tumor immune response [1,28].

The presence of lymphoid aggregates is linked with improved responses to cancer therapies such as standard cytotoxic therapies, vaccine-based treatments, or immune checkpoint blockades. [5,134] Immunologically ‘hot’ tumors, such as melanomas and lung cancers, are thus more amenable to control than ‘cold’ tumors, i.e., tumors with diminished T cell infiltrates, such as GBM. [135,136] This drives modern cancer therapy to investigate how to redirect the TME to attract the right types of immune infiltrates.

Effector arm insufficiency, especially CD8^+^ T cell dysfunction, is a hallmark of inadequate anti-tumor immune response [16]. Four forms of T cell dysfunction—tolerance, anergy, exhaustion, and senescence—have all been reported in cancer microenvironment [17,35,137]. Immune tolerance is a physiological process where the body eliminates self-reacting T cells. Tumor cells, such as GBM, can mimic peripheral tolerance and facilities FasL-mediated deletion of T cells [17]. Tolerance can be enforced by TGF-β and IL-10 secreted by Tregs that are recruited in the TME as well as cancer cells. [17,138,139]. T cell anergy is a T cell hypo-responsive state with low IL-2 production and poor proliferative capacity [17]. It results from the lack of co-stimulation of TCRs through CD28. This is due to the competitive binding of CTLA-4 expressed on Tregs to CD80 and CD86 on the APCs [14,17,139]. Anergic T cells display very little to no effector function, but expression on inhibitory markers is unclear [15]. T cell exhaustion occurs after excessive and continuous stimulation of the TCR and inflammatory cytokines like IFNα/β. Exhausted T cells have diminished proliferative capacity and have poor cytokine production and effector function. However, they express high levels of inhibitory receptors, or immune checkpoints, such as PD-1, CTLA-4, TIM3, LAG3, etc. [15,140]. The degree of T cell exhaustion can vary with tumor types as well. It is more severe in GBM compared to other cancers such as breast, lung, and melanoma [137]. T cell senescence can be distinguished from anergy and exhaustion in their origins and their surface receptors. For example, senescent T cells express fewer CD28 but more NK receptors, whereas exhausted or anergic T cells express more inhibitory receptors such as PD-1 and CTLA-1 (Figure 4). While anergic and exhausted T cells are hyporesponsive and hypofunctional, senescent T cells were considered metabolically active in their physiological or pathological environment despite being in cell cycle arrest (Figure 3). Though both T cell anergy and T cell exhaustion in natural occurrence are considered reversible, T cell senescence was considered irreversible until recently [14].

Targeting dysfunctional effector T cells has revolutionized the paradigm of tumor immunotherapy and immune checkpoint inhibitors set a paramount example [141]. Utilizing tumor dysregulation of immune checkpoint expression in exhausted dysfunctional T cells, immune checkpoint inhibitors were developed to promote protumor immune landscape [141]. Ipilimumab, CTLA-4 inhibitor, the first immune checkpoint inhibitor approved by FDA, was used to treat patients with advanced melanoma and has demonstrated improved survival when given with gp100 melanoma vaccine [142]. PD-1 inhibitors, pembrolizumab and nivolumab, and PD-L1 inhibitors, atezolizumab, durvalumab, and avelumab soon followed showing promising results. Pembrolizumab and nivolumab demonstrated 40–45% objective response rate in melanoma and non-small cell lung cancer [143]. In urothelial bladder cancer, use of PD-1/PD-L1 inhibitors showed overall response rate between 13% and 24% [144]. With metastatic brain disease, the combination of ipilimumab and nivolumab showed 56% intracranial response with 19% complete response from metastatic melanoma, and pembrolizumab have been shown to demonstrate intracranial activity against melanoma and NSCLC metastasis [145]. Continued investigation of the safety and efficacy of these novel immunomodulating drugs are ongoing in various malignancies [146]. However, it has been reported that the presence of functionally aberrant senescent T cells with loss of CD27 and CD28 and gained expression of CD57 cells was associated with resistance to checkpoint inhibitor blockade [8]. Therefore, senescent T cell phenotypes are possible predictive biomarkers for clinical response to checkpoint inhibitor therapy and potential targets to improve checkpoint inhibitor efficacy.

Increased CD8^+^CD28^−^ senescent populations displaying heterogeneous roles have been observed in multiple solid and hematogenous tumors [24]. This immunosuppressive phenotype was initially observed in patients with plasma cell dyscrasia, where increased number of CD8^+^CD28^−^ T cells was present in the bone marrow (i.e., TME) and the amount directly correlated with the suppression of antigen-specific T cell response [123]. Similarly, in patients with lung cancer, the CD8^+^CD28^−^ T cells express elevated Foxp3 and have been shown to play an immunoregulatory role [147]. High levels of CD8^+^CD28^−^ T cells were found in patients with advanced stages of non-small-cell lung cancer. Their numbers declined with resection of the tumor, and the decreased level of CD8^+^CD28^−^ T cells correlates with favorable prognosis in tumor management [148]. In contrast, the CD8^+^CD28^−^ T cell populations in melanoma patients express perforin, where they contribute to anti-tumor immune response [149].

CD8+CD28- T cell senescence is triggered by a variety of biological processes including telomere damage, Treg cells and tumor-associated stresses [150]. Naturally occurring CD4^+^CD25^hi^Foxp3^+^ Treg (nTreg) and tumor-derived γδ Treg cells can induce responder T cell senescence as an immunosuppressive mechanism [127,151]. Senescent T cells induced by Treg cells have phenotypic changes, including expression of SA-β-Gal, downregulation of co-stimulatory molecules CD27 and CD28, and promotion of cell cycle and growth arrest in G0/G1 phase [127,151]. Importantly, CD8^+^CD28^−^ senescent T cells induced by Treg cells have potent regulatory activities [150] and deepen immunosuppression in TME [152]. Therefore, CD8^+^CD28^−^ senescent T cells are important mediators and amplifiers of immunosuppression mediated by Treg cells. The blockage of Treg-induced senescence in responder immune cells is critical in controlling tumor immunosuppression and restoring effector T cell function.

One of the mechanisms for Treg-induced CD8+ T cell immunosenescence is mediated by nuclear kinase ataxia-telangiectasia mutated protein (ATM)-associated DNA damage in responder T cells triggered by glucose competition [150]. MAP ERK1/2 and p38 signaling functionally cooperate with transcription factors STAT1/STAT3 to control Treg-induced senescence in responder T cells [150]. Utilizing these mechanisms, Treg-induced T cell senescence was successfully prevented by inhibiting the DNA damage response and/or STAT signaling in a recent *in vivo* mice study [150]. Another study has shown that senescent T cells are in fact able to regain function by inhibiting the p38 MAPK pathway [153]. Furthermore, human Toll-like receptor 8 (TLR8) signaling can directly target multiple types of tumors and prevent tumor-induced cell senescence through modulation of levels of endogenous secondary messenger cAMP in tumor cells [154].

## 9. CD8^+^CD28^−^ T cells and Glioblastoma

Despite being isolated in the intracranial compartment by the blood brain barrier, GBM, the most common and aggressive primary brain tumor in adults, demonstrates a remarkable level of immunosuppression [155]. Current standard of care for patients with GBM includes surgery, temozolomide chemotherapy, radiotherapy, and corticosteroids, all of which have potent immunosuppressive effects. Tumor cells express surface ligands such as PD-L1 and CD95 (Fas/apoptosis antigen 1) that can lead to T cell suppression via apoptosis and immunosuppressive cytokines like TGF-β, IL-10, and other tolerance factors [139]. Tumor-associate macrophages, modified neutrophils, and Foxp3^+^ Tregs, are also recruited by the tumor to promote its progression [156,157,158].

T cell dysfunctions including tolerance, anergy, and exhaustion have also been well documented in GBM [14,17]. However, despite of promise of checkpoint inhibitors in the treatment of several solid tumors, their therapeutic efficacy in GBM remains to be validated. Phase III clinical trial Checkmate 143 reported that PD-1 monoclonal antibody (nivolumab) monotherapy failed to demonstrate survival benefits compared to bevacizumab in recurrent GBM patients who were previously treated with chemotherapy and radiotherapy [17,69,155]. Ongoing clinical trials are investigating the tolerability and drug toxicity in combination treatment and in patients with newly diagnosed GBM patients as well as recurrent GBM patients. The muted response to immune activators seen thus far highlights to the need for novel strategies to boost immunity to GBM.

The role of T cell senescence in GBM has been reported but is yet to be fully elucidated. The presence of circulating senescent CD4^+^CD28^−^CD57^+^ T cells was correlated with poor prognosis in GBM patients [159]. CD8^+^CD28^−^Foxp3^+^ T cells, which have been found in other cancers to cause APC dysfunction [160], were also identified in TME from patients with GBM [161]. The APCs isolated from these patients displayed dysfunctional phenotype associated with high levels of ILT2, ILT3, and ILT4 and low levels of CD40, CD80, and CD86 [162]. It is speculated that CD8^+^CD28^−^ T cells help sculpt an immunosuppressive environment in a similar fashion in GBM.

The potential pro-tumoral effect of CD8^+^CD28^−^ cells can also be inferred from worse prognosis observed in older GBM patients [163]. Since CD8^+^CD28^−^ cells are derived from the general population of immature CD8^+^CD28^+^ T cells originating from the thymus [99], the production of immature CD8^+^CD28^+^ decreases as thymic involution occurs through aging, but also with cancer [164]. This decrease in immature CD8^+^CD28^+^ cells due to thymic senescence has also been associated with poor outcome in GMB patients [17].

## 10. Implications of CD8^+^CD28^−^ T Cells for the Future of Immunotherapy

Since success of immunotherapy largely relies on addressing effector arm dysfunction, as evident from the success of checkpoint inhibitors, future investigations into new treatment methods should explore strategies to deplete or inhibit regulatory CD8^+^CD28^−^ T cells and reverse T cell senescence as an adjuvant for more effective immunotherapy. There are four main approaches to rejuvenate T cell pools (1) replacement of senescent cells, (2) reprogramming of the senescent cells to be functional, (3) adoptive transfer of proficient T cells, and (4) restoration of naïve T cell pool [165].

Replacement strategies include selectively removing senescent cells from the circulation and then subsequent expansion of memory and effector T cells. Removal of senescent cells is of particular importance, not only due to their own dysfunction but also due to their ability to and spreads senescence in bystander cells [166]. A possible approach for their removal is to promote selective apoptosis in senescent T cells. In a recent study, an engineered peptide that interferes with FOXO4/p53 interaction induced p53-mediated intrinsic apoptosis in senescent fibroblasts and neutralized doxorubicin-induced chemotoxicity *in vivo* [167]. Whether this also can be used in inducing apoptosis of senescent T cell remains to be investigated. Targeting commonly known senescent cell anti-apoptotic pathways such as Bcl-2 and ephrins in senescent T cells is also warranted [168]. Homeostatic expansion in the form of autologous stem cell transplantation has been used to reconstitute functional naïve, memory, and affect T cell pools in both autoimmune diseases and hematologic malignancies [169,170]. More recently, senolytic treatment of amyloid-beta (Aβ) peptide -associated senescent oligodendrocyte progenitor cell in mice with Alzheimer’s disease showed successful selective removal of senescent cells from the plaque, reduced neuroinflammation, lessened Aβ load, and ameliorated cognitive deficits [171]. This is of particular interest as a successful application of senolytic therapy in the CNS pathology, such as GBM. CD8^+^CD28^−^ cells replacement strategies are still in early phases of development, however their successful implementation has the potential to complement the current paradigm of immunotherapies.

Re-programming involves differentiating T cells away from dysfunctional states by enhancing telomerase activity to extend cellular lifespan and preclude replicative senescence [172]. For example, restoring CD28 expression slows replicative senescence in human T cells through increased telomerase activity to increase proliferative potential [173]. Additionally, pharmacological inhibition of SRC homology 2 domain-containing phosphatase-1 (SHP-1), a key regulator of T cell signal transduction machinery, improved TCR/CD28 signaling and successfully improve T cell functions in elderly donors [174]. Aptamers, short, single-stranded DNA or RNA molecules, have been engineered to target immune costimulatory receptors (CD28, OX40 and 4-1BB) and have shown to improve T cell activation and induce antitumor response. Aptamers have the benefits of being chemically synthesized, versatility of targeting motifs, high penetration rate, and ease of neutralization [75]. Re-programming of T cells present the most promising avenue for anti-CD8^+^CD28^−^ therapy with wide selection of potential targets and treatment mechanisms.

Adoptive transfer is to bypass the co-stimulation requirement to re-differentiate pluripotent stem cells into naïve and cytotoxic T cells to fight malignancy [175]. The development of T cell adoptive immunotherapy using the third generation of CAR technology by incorporating the intracellular costimulatory domains to bypass the requirement for CD28 activation is underway. The third generation CARs T cells were investigated in hematological malignancies and xenograft model of solid tumors and have shown preclinical success [176,177,178,179,180]. While other types of immunotherapy can cause systemic side effects, antigen specificity of CAR therapy limits the adverse effect of immunotherapy to its targets, and they are reversible when the target cell is eliminated, or the engraftment of the CAR T cells is terminated [181]. However, its high specificity can be a weakness in heterogeneous tumors with high mutational profiles since CAR therapy can select for cells negative for the targeted antigen [155]. Such was the case for IL-13Rα2 CAR therapy for GBM where recurrence occurred 7.5 months after treatment despite shrinking all lesions by 77–100% [155].

Finally, restoring and maintaining the thymic environment reverse effects of thymic involution. Using bioengineered thymus organoids with the help of growth-promoting factors and cytokines such as IL-21, it has been shown that significant immune restorative function and rejuvenation of the peripheral T cell pools were achieved in murine models [182]. Unfortunately, current understanding of thymic restoration is not complete enough for clinical implementation, and there are still unanswered questions regarding its feasibility in establishing functional naïve T cell production [183,184]. The safe removal of senescent CD8^+^ T cells and restoration or differential induction of functional CD8^+^ cytotoxic T cells would add a promising mechanism to defend host against cancer invasion and fight immunotolerance of malignancy.

There is theoretical concern that reversing the growth arrest by selective blockage of senescent T cells carries a risk of malignancy, which is less of a concern for targeting functionally exhausted T cells [185]. Nevertheless, one may argue that increased theoretical lifetime risk of malignancy is outweighed by the potential immediate benefit of extending the life expectancy, even by a few months to years, in the battle against aggressive cancers, such as GBM with a median overall survival of only 20.6 months. Furthermore, the benefits of targeting both immunosenescence and exhaustion may be more evident with reduced dose requirement for each, thereby reducing risks of drug-associated adverse events. Potential synergistic efficacy to boost immunity against cancer may also be implemented as already seen with GITR stimulation/PD-1 blockade and CTLA-4/ICOS stimulation [74,80,81].

## 11. Conclusions

In summary, functional, tumor specific CD8^+^ cytotoxic T cells drive the adaptive immune response to cancer and are the primary endpoint to most immunotherapies. However, the promise of current cancer immunotherapy has been limited by marked immunosuppression in the TME defined by CD8^+^ T cell dysfunction, especially in immune ‘cold’ cancers, such as GBM. Among the many facets of CD8^+^ T cell dysfunction, including tolerance, anergy, and exhaustion, CD8^+^ T cell senescence, as represented by the CD8^+^CD28^−^ population, is an emerging field and their presence has been described in many cancers. CD8^+^CD28^−^ T cells contribute to tumor immunosuppression and resistance to immunotherapy. Further characterization and investigation into this subset of CD8^+^ T cells will provide novel targets for effective immunotherapy and successful cancer control.

## Figures and Tables

**Figure 1 ijms-20-02810-f001:**
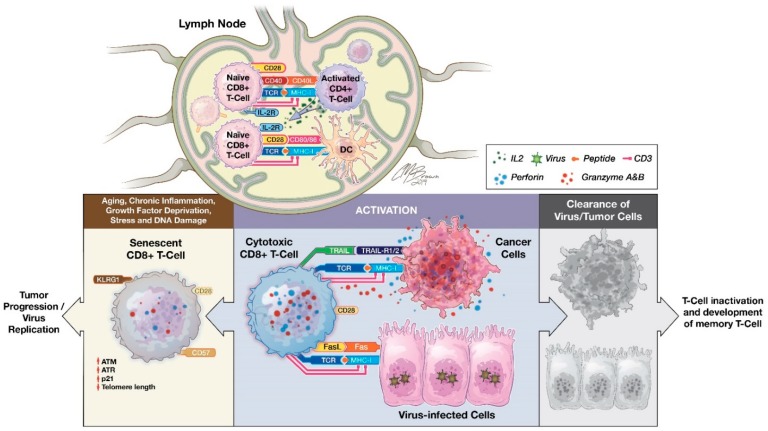
The priming and inactivation of CD8^+^ T cells. The interaction between TCRs and the peptide-MHC complex is the first step toward antigen-induced CD8^+^ T cell activation. This creates a site of extensive contact between T cells and APC, also called immunological synapses, where binding of CD28 on T cells with CD80/CD86 on APCs transduces a pivotal secondary costimulatory signal to complete the priming of naïve CD8^+^ T cells. In addition, CD4^+^ T helper (Th) cells when activated by DCs acquire not only the synapse-composed MHC class II and costimulatory molecules (CD54 and CD80), but also the bystander peptide-MHC-I complex from DC and become CD4^+^ Th-APCs, resulting in direct CD4^+^ T–CD8^+^ T cell interactions and subsequently delivery of CD40L signaling to CD40-expressing CD8^+^ T cells [21]. Furthermore, CD4 Th cells also secrete cytokines, such as IL-2, which promotes the differentiation of naïve CD8^+^ T cells into effector CTLs and memory CD8^+^ T cells. CTLs destroy antigen-specific target cells (such as cancer cells or viral infected host cells) via pathways including granule exocytosis, Fas ligand, and TRAIL-mediated apoptosis leading to tumor control or virus clearance and subsequent physiological T-cell inactivation as well as memory CD8^+^ T-cell formation [33]. Whereas pathological T-cell inactivation or conversion of CTL to CD8^+^ senescent T cells leads to tumor progression and virus replication.

**Figure 2 ijms-20-02810-f002:**
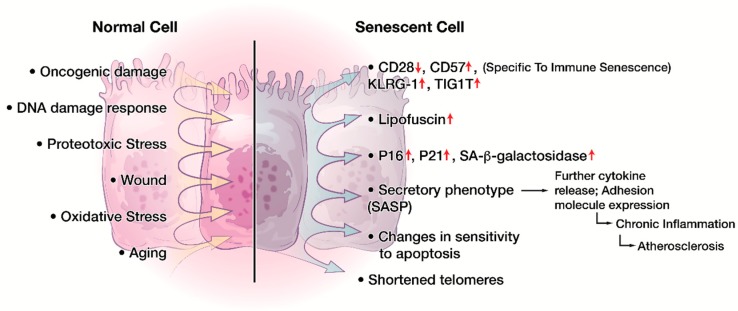
Phenotypical and molecular changes in cellular senescence. A variety of intracellular (DNA damage, oncogenes, etc.) and/or extracellular signals (oxidative stress, chronic inflammation, etc.) can induce cellular senescence. Senescent cells exhibit numerous characteristics including but not limited to cell cycle arrest, increased nuclear p16 and p21 expression, increased lysosomal SA-β-gal activity, shortened telomere, and increased lipofuscin. Senescent cells can also present as a specialized secretory phenotype termed senescence associated secretory phenotype (SASP). Of particular interest, senescent immune cells present with lowered expression of CD28 and CD27 but heightened expression of CD57, KLRG-1, TIGIT, and other NK-cell associated surface receptors.

**Figure 3 ijms-20-02810-f003:**
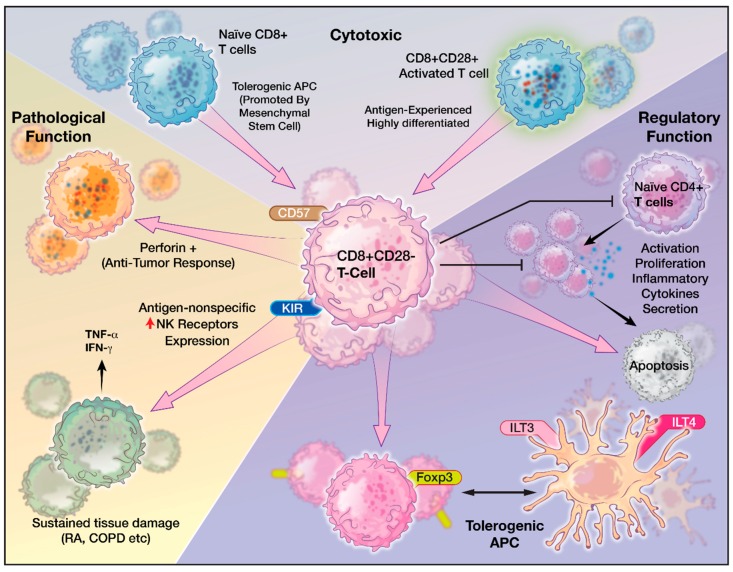
The heterogeneous functions of CD8^+^CD28^−^ T cells. CD8^+^CD28^−^ T cells originate from activated CD8^+^CD28^+^ T cells or from interaction with tolerogenic APCs. CD8^+^CD28^−^ T cells exhibit both cytotoxic and immunoregulatory phenotypes and vary in pathological states such as across different cancer types or inflammatory/autoimmune conditions.

**Figure 4 ijms-20-02810-f004:**
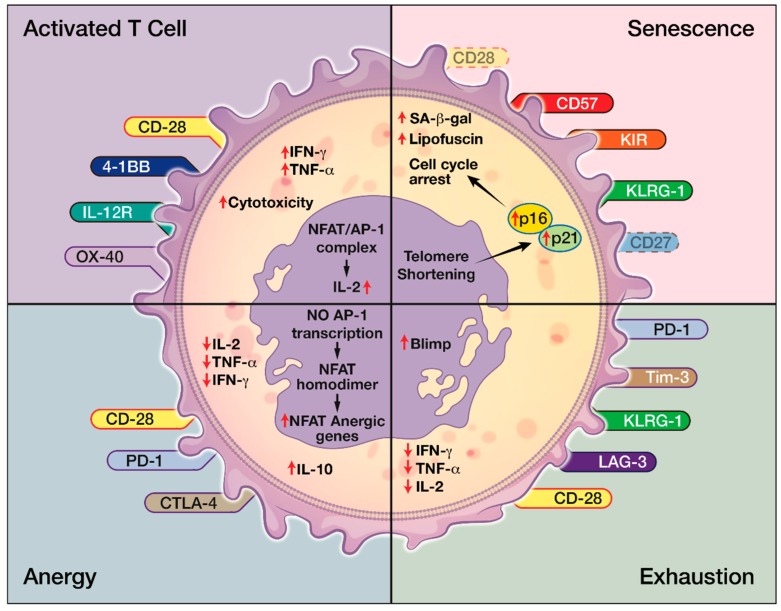
The many facets of C8+ T cell dysfunction. In comparison to activated effector T cells, T cell anergy, exhaustion, and senescence can be distinguished by their unique expression or lack of expression of surface receptors as well as different levels of intracellular cytokines, such as IL-2.

**Table 1 ijms-20-02810-t001:** Clinical trials related to the therapeutic use of CD28 manipulation, such as CAR-T cell therapy and monoclonal antibodies.

Malignancy	Phase	N	Trial Name	Clinical Trial Identifier	Therapeutics	References
Relapsed or Refractory Acute Lymphoblastic Leukemia	1	5	Chimeric Antigen Receptor (CAR)-Modified T Cell Therapy in Treating Patients with Acute Lymphoblastic Leukemia	NCT02186860	Third Gen CAR-T cells containing CD28+CD137	[71]
Glioblastoma	1	17	CMV-Specific Cytotoxic T Lymphocytes Expressing CAR Targeting HER2 in Patients with GBM (HERT-GBM)	NCT01109095	Second Gen CMV-selected CAR-T cells against HER2 containing CD28.zeta signaling domain	[72]
Rheumatoid Arthritis	1/2	18	Safety, Tolerability, Pharmacodynamics and Efficacy Study of TAB08 in Patients with Rheumatoid Arthritis	NCT01990157	TAB08	
Solid Neoplasms	1	38	Dose Escalation Study of TAB08 in Patients with Advanced Solid Neoplasms (TAB08)	NCT03006029	TAB08	[73]
Systemic Lupus Erythematosus	2	730	Safety and Efficacy Study of a Biologic to Treat Systemic Lupus Erythematosus	NCT02265744	Lulizumab pegol (monoclonal antibody against CD28)

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
