# Peer review of "The Evolving Role of CD8+CD28 Immunosenescent T Cells in Cancer Immunology"

_ijms, 2019, doi:10.3390/ijms20112810_

Reviewer 1 Report

The authors provide a review of CD8+ CD28- senescent T cells that encompasses the origin, phenotype and roles of this cell subset, in the context of transplants and disease processes such as autoimmunity, chronic infections, and tumors, followed by a brief summary of the various methods that these cells could be targeted as part of immunotherapy for cancer patients.

Strengths

The article is well written in terms of structure and content. It covers a topic of interest for a sizeable proportion n of immunologists because of the emerging role of immunosenescence in many disease processes, in addition to basic concepts of immune priming and translational research as well. The figure presentations are aesthetically pleasing. Figure 2 and 3 are especially great additions to the paper and are very useful overviews of the current knowledge in this domain. There are good summary sections that describe the role of immunosenescence in different diseases, and the figures that summarize immunosenescent biomarkers or discuss the differences between immunosenescence and other classifications of immunosuppression are excellent. 

Weaknesses

Figure 1 contains general immune concepts that are already known by most immunologists (and is therefore possibly misplaced/redundant/unnecessary in the current specialized review article). I suggest either removing this figure or updating it to include more information, especially information which focuses on immunosenescent pathways and phenomena. The therapeutic section is too brief and requires additional information and references as outlined below.

Specific items to be addressed

·      Line 17, please specify whether you mean naïve or previously activated T cells. NB, typically 3 signals provide optimal priming of naïve T cells.

·      The introduction is clear and concise but should contain the accepted current definition of immunosenescence.

·      Line 60, This theory is critical to your entire thesis so please include the 2 or 3 most important original research citations to support your statement in addition to the current article that is cited here (even though you discuss in more detail later in the review).

·      Figure 1  

o    please include CD86 in addition to CD80 in the DC:T cell activation event

o   Please include other main effector mechanisms that CD8 T cells use to kill tumors (direct lysis by TRAIL, FAS and others, see http://clincancerres.aacrjournals.org/content/21/22/5047). Also mention and cite these additional effector mechanisms on line 104.

o   Is the small boxed area a representation of the LN? If yes, please label this boxed space.

o   In the bottom right hand corner of the large boxed area you could add another arrow from the inactivated T cell towards  labeled images of ‘tumor progression’ and ‘viral replication’ to emphasize the consequences of T cell inactivation

o   Is line 92 supposed to say ‘inactivation’? Otherwise the final sentence of the figure legend doesn’t seem to match the representation of surface molecules in the drawing.

o   update the drawing to include the relative level of CD28 on each of the CD8 T cells shown.

o   In the figure legend, and or pictorially, please mention the mechanisms and functional/biological ramifications of the T cell inactivation step, and clarify whether this is the same or different to senescence. If it’s part of the senescence process please add the word ‘senescence’ to an appropriate part of the picture.

·      Line 136, I highly recommend that you briefly mention the past clinical trials of CD28 superagonists and also if and what advances have been made in the field to reduce the toxicity of this approach.

·      Line 199. A more recent review suggests an alternative interpretation of the papers you cited about NKG2D killing and its TCR (in)dependency in CD8 T cells. https://www.nature.com/articles/cmi2017161. Please check the primary literature carefully and update your statement/citations to include a more comprehensive and accurate description if required.

·      Therapy section:

o   Please include a table that summarizes any clinical trials in this field, citing the NCT numbers, academic publications, CD28 targeting strategy, general results and status of the trials, and any other information you deem useful.

o   Each different type of therapy should be a new paragraph, and expanded to include more details such as how advanced each approach is in terms of clinical development, and include more comparative discussion to allow the reader to understand more about the advantages and disadvantages of each of these approaches.

o   An additional paragraph should be included to compare/contrast the targeting of CD28 to other costimulatory molecules, including those that are modulated by immunosenescence

o   Please cite, discuss and if possible refute the viewpoints made in the paper https://www.nature.com/articles/nri2959.pdf?draft=collection , notably the statement that it would be better to target T cell exhaustion compared to immunosenescence

Author Response

We thank the reviewers for their time and appreciate their constructive feedback. We have reviewed the excellent suggestions from the reviewers and have addressed all their concerns individually as outlined below.

Reviewer 1

The authors provide a review of CD8+ CD28- senescent T cells that encompasses the origin, phenotype and roles of this cell subset, in the context of transplants and disease processes such as autoimmunity, chronic infections, and tumors, followed by a brief summary of the various methods that these cells could be targeted as part of immunotherapy for cancer patients.

Strengths

The article is well written in terms of structure and content. It covers a topic of interest for a sizeable proportion n of immunologists because of the emerging role of immunosenescence in many disease processes, in addition to basic concepts of immune priming and translational research as well. The figure presentations are aesthetically pleasing. Figure 2 and 3 are especially great additions to the paper and are very useful overviews of the current knowledge in this domain. There are good summary sections that describe the role of immunosenescence in different diseases, and the figures that summarize immunosenescent biomarkers or discuss the differences between immunosenescence and other classifications of immunosuppression are excellent. 

Weaknesses

Comment:Figure 1 contains general immune concepts that are already known by most immunologists (and is therefore possibly misplaced/redundant/unnecessary in the current specialized review article). I suggest either removing this figure or updating it to include more information, especially information which focuses on immunosenescent pathways and phenomena. 

Response: We clarified the issue please refer to updated Figure 1.

The therapeutic section is too brief and requires additional information and references as outlined below.

Comment:  Line 17, please specify whether you mean naïve or previously activated T cells. NB, typically 3 signals provide optimal priming of naïve T cells.

Response:We clarified the issue please refer to the abstract.  

Comment:The introduction is clear and concise but should contain the accepted current definition of immunosenescence.

Response:  We clarified the issue please refer to Line 55-56.

Comment:Line 60, This theory is critical to your entire thesis so please include the 2 or 3 most important original research citations to support your statement in addition to the current article that is cited here (even though you discuss in more detail later in the review).

Response: We clarified the issue please refer line 61. 

Figure 1

Comment:please include CD86 in addition to CD80 in the DC:T cell activation event

Response: We clarified the issue please refer to revised figure 1. 

 Comment:Please include other main effector mechanisms that CD8 T cells use to kill tumors (direct lysis by TRAIL, FAS and others, see http://clincancerres.aacrjournals.org/content/21/22/5047). Also mention and cite these additional effector mechanisms on line 104.

Response: We clarified the issue please refer to revised figure 1 and lines 106-108.  

Comment:Is the small boxed area a representation of the LN? If yes, please label this boxed space.

Response: We clarified the issue please refer to revised figure 1.  

Comment:In the bottom right hand corner of the large boxed area you could add another arrow from the inactivated T cell towards labeled images of ‘tumor progression’ and ‘viral replication’ to emphasize the consequences of T cell inactivation

Response:We clarified the issue please refer to revised figure 1.  

Comment:Is line 92 supposed to say ‘inactivation’? Otherwise the final sentence of the figure legend doesn’t seem to match the representation of surface molecules in the drawing. 

Response:We clarified the issue please refer to revised figure 1.  

Comment:update the drawing to include the relative level of CD28 on each of the CD8 T cells shown. 

Response: We clarified the issue please refer to revised figure 1. 

Comment:In the figure legend, and or pictorially, please mention the mechanisms and functional/biological ramifications of the T cell inactivation step, and clarify whether this is the same or different to senescence. If it’s part of the senescence process please add the word ‘senescence’ to an appropriate part of the picture.

Response: We clarified the issue please refer to revised figure 1 and figure 1 legend.  

Comment:Line 136, I highly recommend that you briefly mention the past clinical trials of CD28 superagonists and also if and what advances have been made in the field to reduce the toxicity of this approach. 

ResponseWe clarified the issue please refer to line 139-158.  

Comment:Line 199. A more recent review suggests an alternative interpretation of the papers you cited about NKG2D killing and its TCR (in)dependency in CD8 T cells. https://www.nature.com/articles/cmi2017161. Please check the primary literature carefully and update your statement/citations to include a more comprehensive and accurate description if required. 

Response: We clarified the issue please refer to line 241-243.  

Therapy section:

Comment:Please include a table that summarizes any clinical trials in this field, citing the NCT numbers, academic publications, CD28 targeting strategy, general results and status of the trials, and any other information you deem useful.

Response: We clarified the issue please refer to Table 1. 

Comment: Each different type of therapy should be a new paragraph, and expanded to include more details such as how advanced each approach is in terms of clinical development, and include more comparative discussion to allow the reader to understand more about the advantages and disadvantages of each of these approaches.

Response:We clarified the issue please refer to line 439-507.  

Comment:An additional paragraph should be included to compare/contrast the targeting of CD28 to other costimulatory molecules, including those that are modulated by immunosenescence

Response:We clarified the issue please refer to line 167-177.  

Comment:Please cite, discuss and if possible refute the viewpoints made in the paper https://www.nature.com/articles/nri2959.pdf?draft=collection , notably the statement that it would be better to target T cell exhaustion compared to immunosenescence

Response: We clarified the issue please refer to line 492-501.

Reviewer 2 Report

In the manuscript the authors give a comprehensive and well-structured overview about the functional and phenotypic properties of the CD8+CD28+ senescent T cell subset, their interplay with other populations and their implication for immunotherapy.

The review is clear written and address all important points to get insights into the role of these underestimated cells. Figures are well done and help to follow important aspects. 

Author Response

Reviewer 2

Comments and Suggestions for Authors

Comment:In the manuscript the authors give a comprehensive and well-structured overview about the functional and phenotypic properties of the CD8+CD28+ senescent T cell subset, their interplay with other populations and their implication for immunotherapy.The review is clear written and address all important points to get insights into the role of these underestimated cells. Figures are well done and help to follow important aspects. 

Response: We thank the reviewer for the positive comments.

Round  2

Reviewer 1 Report

The authors have successfully and comprehensively addressed the original comments. I recommend publication after correction of the remaining grammar and spelling in the newly added sections of the manuscript.

Author Response

Comment: The authors have successfully and comprehensively addressed the original comments. I recommend publication after correction of the remaining grammar and spelling in the newly added sections of the manuscript.

Response: We have gone though the entire manuscript and have addressed all the grammar/spelling issue.